# Are Dual-Phase ^18^F-Fluorodeoxyglucose PET-mpMRI Diagnostic Performances to Distinguish Brain Tumour Radionecrosis/Recurrence after Cranial Radiotherapy Usable in Routine?

**DOI:** 10.3390/cancers16183216

**Published:** 2024-09-21

**Authors:** Axel Cailleteau, Ludovic Ferrer, Delphine Geffroy, Vincent Fleury, Paul Lalire, Mélanie Doré, Caroline Rousseau

**Affiliations:** 1Department of Radiotherapy, ICO René Gauducheau, 44800 Saint Herblain, France; melanie.dore@ico.unicancer.fr; 2Department of Nuclear Medicine, ICO René Gauducheau, 44800 Saint Herblain, France; ludovic.ferrer@ico.unicancer.fr (L.F.); paul.lalire@ico.unicancer.fr (P.L.); caroline.rousseau@ico.unicancer.fr (C.R.); 3Nantes Université, Univ Angers, INSERM, CNRS, CRCI2NA, F-44000 Nantes, France; 4Department of Radiology, ICO René Gauducheau, 44800 Saint Herblain, France; delphine.geffroy@ico.unicancer.fr

**Keywords:** dual-phase FDG, PET-MRI, brain radionecrosis, cerebral tumour recurrence, cranial radiotherapy

## Abstract

**Simple Summary:**

Distinguishing radionecrosis from local recurrence after cranial radiotherapy remains a challenging diagnostic dilemma due to their overlapping clinical manifestations. Although new MRI techniques and metabolic imaging have been developed, diagnostic performance remains variable. Recently, our institution acquired a PET-MRI, enabling us to investigate the diagnostic performance of multiparametric MRI when used in dual-phase ^18^F-FDG PET-mpMRI to enhance diagnostic accuracy.

**Abstract:**

Brain metastases or primary brain tumours had poor prognosis until the use of high dose radiotherapy. However, radionecrosis is a complex challenge in the post-radiotherapy management of these patients due to the difficulty of distinguishing this complication from local tumour recurrence. MRI alone has a variable specificity and sensibility, as does PET-CT imaging. We aimed to investigate the diagnostic performance of dual-phase ^18^F-FDG PET-mpMRI to distinguish cerebral radionecrosis from local tumour recurrence after cranial radiotherapy. A retrospective analysis was conducted between May 2021 and September 2022. Inclusion criteria encompassed patients with inconclusive MRI findings post-radiotherapy and history of cerebral radiotherapy for primary or metastatic brain lesions. Lesions are assessed qualitatively and semi-quantitatively. The gold standard to assess radionecrosis was histopathology or a composite criterion at three months. The study evaluated 24 lesions in 23 patients. Qualitative analysis yielded 85.7% sensitivity and 75% specificity. Semi-quantitative analysis, based on contralateral background noise, achieved 100% sensitivity and 50% specificity. Moreover, using contralateral frontal lobe background noise resulted in higher performances with 92% sensitivity and 63% specificity. Stratification by lesion type demonstrated 100% sensitivity and specificity rates for metastatic lesions. The diagnostic performance of dual-phase ^18^F-FDG PET-mpMRI shows promising results for metastatic lesions.

## 1. Introduction

Brain metastases from solid tumours or primary brain tumours used to have a poor prognosis. [1,2]. However, management of both conditions has improved: advancements in stereotactic brain radiotherapy [3] and enhanced intracranial efficacy of drugs in medical oncology have led to better survival rates [4,5,6,7]. Nevertheless, higher doses administered in radiotherapy increase the risk of complications such as cerebral radionecrosis [8,9]. Distinguishing cerebral radionecrosis from local tumour recurrence poses a tough diagnostic dilemma due to their overlapping clinical manifestations (seizures, increased intracranial pressure, focal deficits) and analogous radiological features (contrast enhancement, oedema) [10]. Despite its widespread use, conventional magnetic resonance imaging (MRI) has demonstrated an inconsistent diagnostic performance [11].

While definitive diagnosis often relies on histopathology, this approach is limited by its invasiveness in cerebral lesions. Consequently, serial MRIs over extended intervals are usually recommended, depending on temporal evolution to guide clinical decision-making. However, this approach can lead to prolonged uncertainty and heightened patient anxiety and delay any targeted treatment [12].

Multiparametric MRI (mpMRI) techniques have been developed to enhance diagnostic accuracy and improve patient care, including dynamic perfusion sequences. Some series have reported remarkable sensitivities of 100% and specificities of 95% using a relative cerebral blood volume (rCBV) cutoff of 2.1. Extensive literature substantiates its utility in discriminating recurrences, exhibiting a sensitivity ranging from 78.92% to 100% and a specificity ranging from 71.58% to 100%, using threshold values varying between 1.54 and 2.1 [13,14,15,16]. Nevertheless, this sequence remains susceptible to magnetic susceptibility artefacts and, so, occasionally non-exploitable in the presence of haemorrhagic alterations or proximity to bone structures. Furthermore, the diagnostic threshold for rCBV manifests variability across different imaging systems, thereby introducing complexities into its routine application.

Metabolic imaging has been explored extensively, particularly the commonly used 18F-fluorodeoxyglucose (^18^F-FDG) positron emission tomography and computer tomography (PET/CT). However, the reported sensitivities range widely from 36% to 95%, with specificities varying from 50% to 100% [17]. Horky et al. introduced an innovative approach to improve the diagnostic performance of ^18^F-FDG PET/CT. By employing a sequential acquisition strategy involving both early and delayed imaging phases, they calculated a retention index with a cutoff value of 0.19. This approach yielded an impressive sensitivity of 95% and specificity of 100% in distinguishing between cerebral radionecrosis and local recurrence [18].

Our institution recently acquired a PET-MRI, which allowed us to investigate the diagnostic performances of mpMRI when coupled with dual-phase ^18^F-FDG PET imaging in patients with suspected brain radionecrosis or local recurrence.

## 2. Materials and Methods

### 2.1. Patient Selection

We retrospectively collected patient data referred for dual-phase ^18^F-FDG PET-mpMRI imaging at our centre between May 2021 and September 2022 to distinguish cerebral radionecrosis from intracranial recurrence following external radiotherapy. Inclusion criteria were patients over 18 years old who had undergone a dual-phase ^18^F-FDG PET-mpMRI due to inconclusive findings on prior MRI regarding the differentiation between radionecrosis and recurrence. Additionally, patients needed to have undergone cerebral radiotherapy for a primary brain lesion or one or more metastatic lesions at least three months prior. Exclusion criteria were a recent cerebral biopsy within the last ten days preceding the ^18^F-FDG PET-mpMRI, the impossibility of following the patient for over three months, and patient opposition to data collection. The gold standard reference was histopathology when available; otherwise, a composite criterion was applied to determine recurrence or radionecrosis at three months. This criterion involved the absence of new therapeutic intervention within three months, the absence of a decision of supportive care, image findings at three months (using mpMRI alone) confirming the ^18^F-FDG PET-mpMRI diagnosis, absence of death, and absence of distant intracranial recurrence from the lesion under study.

### 2.2. ^18^F-FDG PET-mpMRI Protocol

Patients were required to fast for 6 h and were placed in a sensory-deprived, dark environment without stimuli for 15 min before injecting ^18^F-FDG. A 3 MBq/kg dose was administered, followed by continued sensory deprivation in darkness until the early acquisition phase. Patients were positioned supine on PET-MRI 3 Tesla (Siemens Biograph mMR 32-Channel Head Coil). The early acquisition phase was initiated 30 min after the FDG injection, including T1 MR-based attenuation correction (MRAC), PET acquisition, 3D T2 space fluid-attenuated inversion recovery (FLAIR), 3D T1 volumetric interpolated breath-hold examination (VIBE), susceptibility-weighted imaging (SWI), diffusion-weighted imaging (DWI) with a b-value of 1000 s/mm^2^ and apparent diffusion coefficient (ADC) mapping, cerebral perfusion imaging with relative cerebral blood volume (rCBV) mapping, gadolinium injection, and a final 3D T1 VIBE scan. A late acquisition phase, 4 h post-injection, including T1 MRAC, PET acquisition, and 3D T2 space FLAIR, followed.

### 2.3. Image Analysis

Image analysis was performed using two modalities:

A qualitative assessment was conducted in real-world conditions based on joint interpretation by senior radiologists and nuclear medicine physicians. This analysis was systematically carried out when patients underwent ^18^F-FDG PET-mpMRI.

Morphological analysis on various sequences was applied, particularly post-gadolinium contrast enhancement: typically showing peripheral enhancement with central hyposignal in brain metastases, the extent of peripheral FLAIR hypersignal (often less extensive in brain metastases than in recurrences), identification of central hyposignal areas on SWI, sequence suggestive of central necrosis, semi-quantitative analysis of perfusion mapping reflecting neovascularisation with the calculation of the rCBV ratio in the pathological area compared to the healthy contralateral area and comparison with prior MRIs to assess lesion progression. The nuclear medicine physician then interpreted the area of uptake in relation to the suspicious area described by the radiologist to obtain a qualitative analysis of the ^18^F-FDG PET scan: persistent hyperfixation on delayed imaging is indicative of potential recurrence.

Semi-quantitative analysis was conducted using external software (slicer 5.2.1) following digital imaging and communications in medicine (DICOM) data export. An oncology radiation therapist, blinded to the patient’s medical history, delineated the target lesion on the MRI 3D T1 with contrast enhancement and replicated this delineation on the exact opposite side (in the contralateral healthy cerebral cortex). Another region of interest (ROI) was delineated in the contralateral superior frontal gyrus, resulting in three delineations per patient. Thanks to the co-registration of PET-MRI, the delineations on MRI automatically calculate the SUVmax of the ROI.

Subsequently, the ratio described by Horky was calculated, comparing the lesion with the contralateral normal brain or the contralateral frontal cortex [18]. So, two different ratios were obtained.

### 2.4. Statistical Analysis

Statistical analysis was performed with XLSTAT^®^ version 2022.1.2.1244. All sensitivities and specificities for semi-quantitative analysis were calculated using a cutoff of 0.19 as previously described by Horky et al. [18]. Confidence intervals at 95% (IC95%) were calculated for both sensitivity and specificity.

## 3. Results

Within our institution, a total of 23 patients underwent ^18^F-FDG PET-mpMRI for this specific indication between May 2021 and September 2022, resulting in the assessment of 24 lesions. Among these, 22 lesions were evaluated qualitatively (2 were lost to follow-up for the gold standard), 21 underwent semi-quantitative analysis (1 was non-interpretable, and 2 were lost to follow-up). Among the lesions, 10 were primary brain tumours (9 glioblastomas and 1 oligodendroglioma), while the remaining 14 were metastases from extracerebral tumours (Table 1). Concerning the gold standard, pathology was available in 18.2% of cases in our cohort.

In qualitative analysis, out of the 22 evaluable lesions, the sensitivity was 85.7% (IC 95% [0.657–1]), and the specificity was 75% (IC 95% [0.333–1]). In semi-quantitative analysis based on contralateral background noise, the sensitivity was 100% with a specificity of 50% (IC 95% [0.152–0.848]). When based on the background noise of the contralateral frontal lobe, the sensitivity was 92% (IC 95% [0.779–1]), and specificity was 63% (IC 95% [0.189–1]).

When analysing the lesions separately based on primary or metastatic tumour status, for the 10 primary brain lesions, the qualitative analysis yielded a sensitivity of 80% (IC 95% [0.275–1]) and a specificity of 60% (IC 95% [0.009–1]), while both semi-quantitative methods resulted in a sensitivity of 100% and specificity of 20% (IC 95% [0–0.626]). For the 12 brain metastases, the qualitative analysis demonstrated a sensitivity of 89% (IC 95% [0.658–1]) and a specificity of 100%. The semi-quantitative analysis based on contralateral background noise achieved a sensitivity and specificity of 100%, while using the contralateral frontal lobe background noise resulted in a sensitivity of 89% (IC 95% [0.658–1]) and a specificity of 100%.

Table 2 presents the diagnosis initially determined by routine MRI that subsequently led to using ^18^F-FDG PET-mpMRI for a second opinion. The “Qualitative” column represents the diagnosis made by the dual interpretation of the nuclear medicine physician and radiologist using ^18^F-FDG PET-mpMRI. The “Contralateral nIR” and “Frontal nIR” columns represent the results of the Horky index calculation conducted by a blinded radiotherapist. The “Gold Standard” column represents the definitive diagnosis determined by histopathology or the composite criterion described in the Methods section. Radionecrosis/recurrence illustrations are represented in Figure 1 and Figure 2.

## 4. Discussion

Cerebral radionecrosis remains a diagnostic challenge in the increased number of indications for the high-dose stereotactic cerebral radiotherapy era. The implications of this diagnosis on patient management and quality of life are fundamental, highlighting the critical importance of developing imaging tools to enhance diagnostic accuracy. The combined impact of PET-FDG/CT and mpMRI is well known. Our study is the first to explore the role of these two modalities combined, precisely sequential ^18^F-FDG PET-mpMRI, in differentiating between cerebral radionecrosis and local recurrence, both in primary brain tumours and brain metastases, where the diagnostic performance appeared highly promising in the semi-quantitative data (sensitivity 100% and specificity 100%).

In the literature, Pyatigorskaya et al. examined a similar topic without incorporating a delayed acquisition phase and high-grade gliomas. This study reported a diagnostic accuracy 0.957. However, these excellent results were not replicated in our glioma patient, demonstrating a specificity of only 60% or even 50% based on contralateral background noise. This difference can be attributed to the gold standard we use compared to that of Pyatigorskaya et al. [19]. Indeed, in their study, their gold standard was solely the absence of retreatment at three months. Our composite criterion could thus detect more false positives for radionecrosis, with more patients being considered in relapse at three months (progressing MRI, new therapeutic line, palliative care on progression).

No previous studies have reported on sequential ^18^F-FDG PET-mpMRI in the context of brain metastases. Nonetheless, the findings of three studies [18,20,21] support the favourable outcomes achieved with ^18^F-FDG but with a CT scan. Otman et al. demonstrated the utility of ^18^F-FDG PET/CT [20]. This study compared the diagnostic performance between dual point and delayed acquisition. The specificity was similar (0.92) but with better sensitivity for delayed only (0.83 vs. 0.63). Our combined analysis with mpMRI possibly allowed better sensitivity thanks to the perfusion sequences associated. Matuszak et al. also reported positive outcomes with delayed acquisition [21] and confirmed the improvement of diagnosis accuracy of the delayed acquisition compared to standard ^18^F-FDG PET/CT. Regarding the use of ^18^F-FDG, its advantage lies in its widespread availability across nuclear medicine centres. A recent meta-analysis did not demonstrate the superiority of one tracer over another (DOPA, MET, FET, FDG) [22], although there was a trend for better accuracy with FDG. While most studies use [^18^F]DOPA [23] with favourable outcomes, current recommendations advocate using amino acids in this context [24,25,26].

The advantage of ^18^F-FDG PET-mpMRI lies in its capacity to integrate these novel techniques into a single examination, continually improving diagnostic accuracy for our patients. In clinical practice, this co-registration also enables the precise delineation of ROI using MRI to obtain the SUVs of these ROIs, thus overcoming the registration limitations between a PET scan and an MRI acquired at different times. Additionally, one of the advantages is the ability to detect a recurrent nodule within a large area of radionecrosis more easily (Figure 3). 

In the future, the addition of new mpMRI sequences, such as Amide Proton Transfer (APT) [27,28], or Arterial Spin Labelling Perfusion MRI (ASL) [29,30], could be easily associated with ^18^F-FDG PET-mpMRI.

The main limitations of our study, apart from its retrospective and single-centre nature, are primarily related to the heterogeneity between primary and secondary brain tumours, leading to a limited number of evaluated lesions in each subgroup, which challenges the formal interpretation of results. Additionally, inter-observer variability was not assessed, although the semi-quantitative approach is less likely to be influenced by differing lesion delineations. Indeed, the ROI for the target lesion was easily delineated on DICOM thanks to MRI-PET co-registration to locate the suspected area of necrosis or recurrence. This ROI was then automatically copied and pasted to the contralateral cerebral coordinates. Finally, the same ROI was anatomically copied and pasted to the contralateral frontal lobe. Inter-observer reproducibility was therefore not assessed. Another significant limitation is our low biopsy rate for confirming the diagnosis, which may call our results into question. Nevertheless, proposing such an invasive procedure to confirm the diagnosis is always challenging. However, a study like that of Horky et al. [18] reported a 64% biopsy rate. It is important to note that this study now serves as a reference for establishing the threshold, and we base our approach on it. Consequently, our physicians place greater confidence in the dual-phase PET results, and likely even more so when combined with MRI, as in our study. Therefore, they are less likely to resort to biopsy to confirm the diagnosis. Additionally, we have chosen a very broad composite criterion that accounts for various scenarios, allowing for accurate patient reclassification. Further studies could be justified to confirm this or to investigate the impact of the Horky-defined cutoff value of 0.19.

## 5. Conclusions

This study presented promising results regarding the utility of dual-phase ^18^F-FDG PET-mpMRI to distinguish cerebral radionecrosis from local tumour recurrence following cranial radiotherapy. Future studies could encompass larger and more diverse cohorts, examine inter-observer variability factors, and assess the potential impact of Horky’s defined threshold values. In our centre, a cohort study exploring the implementation of prospective pre-therapeutic ^18^F-FDG PET-mpMRI is ongoing.

## Figures and Tables

**Figure 1 cancers-16-03216-f001:**
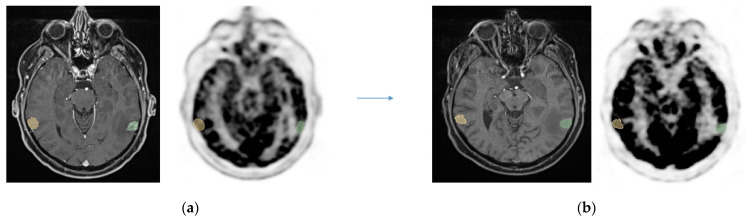
A patient with a lung cancer metastatic cerebral lesion for which multiparametric MRI alone could not differentiate radionecrosis from local recurrence. Early acquisition (**a**) ^18^F-FDG PET-mpMRI showed a lesion in green ROI with SUVmax = 5.69 and contralateral ROI in yellow with SUVmax = 8.66. Late acquisition (**b**) showed lesion SUVmax = 5.04, contralateral SUVmax = 10.53. The Horky ratio is calculated at −0.27, which supports radionecrosis. This diagnosis has been confirmed with patient follow-up (imaging stability over 3 years).

**Figure 2 cancers-16-03216-f002:**
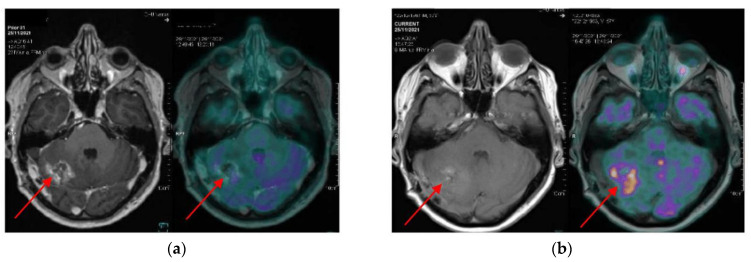
A patient with non-small cell lung cancer with a single metastatic brain lesion was treated with surgical excision followed by radiotherapy on a tumour bed. Due to clinical symptoms (cerebellar syndrome), an ^18^F-FDG PET-mpMRI was performed. The late acquisition (**b**) showed significant FDG retention within the tumour compared to the early acquisition (**a**), with a retention index of 0.87, indicating a local recurrence. Therefore, this patient underwent surgery, and the histopathological results confirmed this hypothesis.

**Figure 3 cancers-16-03216-f003:**
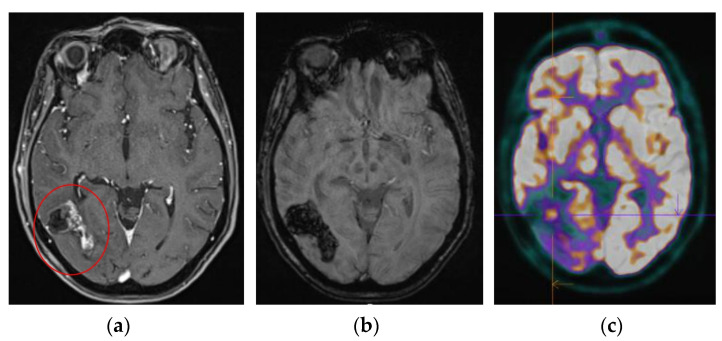
Patient treated for a glioblastoma (radiotherapy–temozolomide) with a large area of radiation necrosis (red circle, (**a**)). Through ^18^F-FDG PET-mpMRI co-registration, the patient was reclassified as having a recurrence within this area of radiation necrosis. Diffusion and perfusion sequences were not interpretable due to haemorrhagic artifacts ((**b**)—SWI sequence); however, thanks to PET co-registration, a hypermetabolism was identified (**c**).

**Table 1 cancers-16-03216-t001:** Patients’ and lesions’ characteristics.

	N = 23
Sex	
Male	10
Female	13
Median age (years, range)	60.5 (29–74)
Pathological analyses	N = 24
Glioblastoma	9
Oligodendroglioma	1
Lung adenocarcinoma	8
Small cell lung cancer	2
Breast cancer	1
Melanoma	3
Treatment	N = 24
Post-surgery	10
RT alone	14
Lesions’ location	N = 24
Sus-tentorial	19
Sub-tentorial	3
Meningeal contact	10
Concomitant therapy, n (%)	N = 14 (60.9)
Immunotherapy	7
Alectinib	1
Temozolomide	6
Re-irradiation, n (%)	N = 8 (34.8)
Whole brain	2
IMRT (partial brain)	3
Stereotactic RT	3

**Table 2 cancers-16-03216-t002:** Summary of ^18^F-FDG PET-mpMRI analysis with the discordance between pre MRI analysis and gold standard.

Patient	Histology	Diagnosis Evocated at Initial mpMRI	^18^F-FDG PET-mpMRI			Gold Standard
			Qualitative	nIR Contralateral	nIR Frontal	
1	Glioblastoma	Relapse	RN	RN	RN	RN
2	NSCL mets	RN	RN	RN	RN	RN
3	Glioblastoma	Equivocal	RN	RN	RN	Relapse
4	Oligodendroglioma	Equivocal	Relapse	Relapse	Relapse	Relapse
5	NSCL mets	Equivocal	Relapse	Relapse	Relapse	Relapse
6	Glioblastoma	RN	Relapse	Relapse	RN	RN
7	NSCL mets	Equivocal	Relapse	Relapse	Relapse	Relapse
8	Melanoma mets	RN	RN	RN	RN	RN
9	Melanoma mets	Equivocal	RN	RN	RN	RN
10	Breast mets	Equivocal	RN	RN	Relapse	RN
11	Glioblastoma	Equivocal	RN	RN	RN	RN
12	Glioblastoma	RN	Relapse	RN	Relapse	Relapse
13	NSCL mets	RN	RN	RN	RN	RN
14	NSCL mets	Equivocal	RN	RN	RN	RN
15	Glioblastoma	RN	RN	RN	RN	Relapse
16	Glioblastoma	Equivocal	RN	RN	RN	RN
17	Glioblastoma	Relapse	Relapse	RN	RN	Relapse
18	Melanoma mets	Equivocal	Relapse	Relapse	Relapse	Relapse
19	SCLC mets	Equivocal	RN	RN	RN	RN
20	NSCL mets	Relapse	Relapse	RN	RN	RN
21	NSCL mets	Relapse	RN	RN	RN	RN

## Data Availability

Data are available upon request to the corresponding author.

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
