# Peer review of "Are Dual-Phase 18F-Fluorodeoxyglucose PET-mpMRI Diagnostic Performances to Distinguish Brain Tumour Radionecrosis/Recurrence after Cranial Radiotherapy Usable in Routine?"

_cancers, 2024, doi:10.3390/cancers16183216_

Round 1

Reviewer 1 Report

Comments and Suggestions for Authors

This study investigates the diagnostic performance of dual-phase 18F-FDG PET-mpMRI in distinguishing cerebral radionecrosis from local tumor recurrence after cranial radiotherapy.

My comments:

Material and Methods:

Image Analysis:  

The authors do not provide information on how the images were analyzed by the two evaluators. It’s unclear if they  reached a consensus, if there was a third evaluator in case of disagreement, or how potential discrepancies were resolved.

The acronym DICOM is not spelled out.

Experience level of the oncology radiation therapist performing the semi-quantitative analysis?

There's no mention of any measures taken to ensure consistency in ROI delineation across patients.

Discussion:

The authors should address whether their sample size is sufficient for generalizing the results to other populations. If the sample size is deemed small or not fully representative of the target population, this should be clearly stated as a limitation.

Reviewer 2 Report

Comments and Suggestions for Authors

The topic is very important. However, the following points need to be addressed.

The first sentence of the abstract seems incorrect. Are there any references to support it? The composite criterion appears inadequate for accurately distinguishing between true tumor progression and radiation necrosis, especially considering that the gold standard, pathology, was only available in 18.2% of cases in the cohort. Therefore, the findings are not entirely convincing.

Please provide more details about the statistical analysis. The determination of the cutoff in the current study requires further detailed explanation.

Round 2

Reviewer 1 Report

Comments and Suggestions for Authors

This revised version fulfilled my raised suspicion.

Reviewer 2 Report

Comments and Suggestions for Authors

I am satisfied with the point-to-point response.